# Design of the topology
# for contrastive visual-textual alignment

## Abstract

Cosine similarity is the common choice for measuring the distance between the feature representations in contrastive visual-textual alignment learning. However, empirically a learnable softmax temperature parameter is required when learning on large-scale noisy training data. In this work, we first discuss the role of softmax temperature from the embedding space's topological properties. We argue that the softmax temperature is the key mechanism for contrastive learning on noisy training data. It acts as a scaling factor of the distance range (*e.g.* [-1, 1] for the cosine similarity), and its learned value indicates the level of noise in the training data. Then, we propose an alternative design of the topology for the embedding alignment. We make use of multiple class tokens in the transformer architecture; then map the feature representations onto an oblique manifold endowed with the negative inner product as the distance function. With this configuration, we largely improve the zero-shot classification performance of baseline CLIP models pre-trained on large-scale datasets by an average of 6.1%.

## 1 Introduction

**Development of Contrastive Alignment:** Learning visual and textual feature representations that are semantically aligned in their embedding space is an ordinary problem in the vision-language cross-modal tasks (Frome et al., 2013; Karpathy & Fei-Fei, 2015; Romera-Paredes & Torr, 2015; Wang et al., 2016; Faghri et al., 2017; Xian et al., 2016). In early works that employ feature representations from deep neural networks, *e.g.* Frome et al. (2013), the alignment is often achieved by a fundamental metric learning approach with the hinge rank loss. That is, the similarity between a visual feature vector $\boldsymbol{u}$ and a textual feature vector $\boldsymbol{v}$ is calculated as $\boldsymbol{u}^T W \boldsymbol{v}$, where $W$ are the learnable weight parameters. Thanks to the revolutionary advances in computational power, we can now achieve this in a more effective and practical approach termed contrastive learning, where we align quantities of positive samples and push their negative samples away simultaneously in a large mini-batch (Radford et al., 2021; Singh et al., 2022; Jia et al., 2021; Pham et al., 2021; Yuan et al., 2021).

**Cosine Similarity:** The common choice of the distance measure between an image-text pair for the contrastive learning algorithm is the `Cosine Similarity` (in both uni-modal Chen et al. (2020a); Caron et al. (2020); Chen et al. (2020b) and cross-modal Radford et al. (2021); Jia et al. (2021); Singh et al. (2022) scenarios). Mathematically, the `Cosine Similarity` computes the inner product value between feature representation vectors mapped onto the unit spherical embedding space. Such embedding space has two properties that are considered advantageous in aligning visual and textual feature representations. First, calculating the inner product consumes low computational resources during both forward and backward propagation. Second, we have a proper definition of uniformity on the sphere, where uniformly distributed feature representations preserve the data's maximal information, optimizing the contrastive loss.

**Learnable Temperature Trick:** Embarrassingly, the original version of the contrastive loss using the `Cosine Similarity` is challenging to train. Therefore, a learnable softmax temperature is prepended and continuously updated through gradient descent, along with the training progress in practice (Wu et al., 2018; Radford et al., 2021). However, this trick has at least two drawbacks. Firstly, a large temperature value is

numerically unstable for the back-propagation, especially for low-bit precision computation. Practically, an upper limit of 100.0 is often used to prevent numerical overflow. Second, we observe that the model acquires a proper scaling for the distance range earlier than achieving a good alignment. We consider that optimizing the temperature parameter delays the learning progress (See Section 4.3 ).

**"Equilibrium" for Noisy Samples:** Now we discuss the mechanism behind the learnable temperature trick. Since the data for large-scale contrastive alignment are internet-collected noisy image-text pairs, we often find pairs of semantically related images and texts labeled as "negative" and verse visa, which we term "semantic ambiguity". Because of the ambiguity, it is impossible to achieve the perfect alignment and uniformity conditions of sample embeddings for the system. More specifically, during the training, the false negative samples are pushed away from each other (repulsion), while the false positive samples are pulled together (attraction). Consequently, the system will gradually find an equilibrium when the noisy samples' gradients for attraction and repulsion are neutralized. In other words, we say the training progress is *converged* under the given hyper-parameters. To be more concrete, owing to the fact that the gradient is eventually back-propagated from the difference between the positive and negative distances. Given sufficient model capacity, the numerical values between the distances of positive and negative pairs of samples will be optimized to fit the noisy level of the dataset.

For instance, if there is a reasonable amount of false negative samples, the model would learn a minor positive similarity for not being punished too hard when encountering false negative samples in another mini-batch. On the other hand, semantically similar samples would agglomerate (learn larger positive similarity) due to the restriction of the triangular inequality (or a "relaxed" version, see Section 3.2). Finally, the model reaches the equilibrium of compromised positive and negative distances, which minimizes contrastive loss under semantic ambiguity.

**Temperature Scales the Distance Range:** Here, the distance range becomes problematic for reaching equilibrium. Remind that the contrastive loss is implemented with the combination of softmax and cross-entropy, which makes the required numerical values for equilibrium exponentially larger than the similarity defined within $[-1, 1]$. Therefore, we are in need of the learnable softmax temperature to expand the distance range to $[-\tau, \tau]$. For instance, the officially released CLIP model Radford et al. (2021) has a glancing similarity of $0.3 \sim 0.5$ and $0.1 \sim 0.3$ for positive and negative pairs of samples, respectively. While the learned temperature is approaching 100.0, indicating the equilibrium distances are $30 \sim 50$ and $10 \sim 30$ for positive and negative pairs of samples, respectively.

**Contributions of This Work:** In this work, we alternatively design the topology for embedding vectors and its endowed distance function. Motivated by the utilization of Riemannian geometry for visual tasks and the class token in transformer architectures, we propose a relatively simple solution to address the aforementioned out-of-range equilibrium problem. Our contributions can be summarized as follows:

1. We argue that the learnable softmax temperature is the key mechanism for learning on noisy training data. We reveal that the temperature is essentially a scaling factor for the distance range, which indicates the noise level of the dataset in the contrastive visual-textual alignment. We also observe that the model learns a proper temperature before representations.

2. We unscramble four neglected properties of the embedding space. Following that, we tackle the out-of-range equilibrium problem by employing an oblique manifold with the inner product distance as the topology for embeddings.

3. We implement the oblique topology with multiple class tokens of the transformer architecture. In the larger scale experiment, we have learned a ViT-B/16-based CLIP model that outperforms the baseline model by an average of 6.1% in zero-shot classification tasks.

## 2 Preliminary

**Notations:** We start with notation and review mathematical expressions of the basic building blocks used in our analysis. In this work, we denote scalars by italic letters, *e.g.*, $n, m, B, D \in \mathbb{R}$, and denote vectors

and higher-order tensors by boldface letters, *e.g.,* $\mathbf{x} = [x_0, x_1, \ldots, x_{n-1}]^\top \in \mathbb{R}^n$ and $\mathbf{Y} \in \mathbb{R}^{N \times D}$. We denote sets by calligraphic letters, *e.g.,* $\mathcal{U} = \{\boldsymbol{U}_1, \boldsymbol{U}_2, \ldots\}$. We also employ italic letters to define functions, with subscripts denoting their parameters, *e.g.,* $f_\theta(\cdot)$. The operation $\| \cdot \|_p$ denotes the $\ell_p$ norm of a vector and $| \cdot |$ denotes the absolute value of a scalar. For any integer $K$, we use $[K]$ to denote the set of integers from 1 to $K$.

**Visual-Textual Pre-trained Model:** Given a set of semantically related image-text pairs $\mathcal{S} = \{(\boldsymbol{U}_1, \boldsymbol{V}_1), (\boldsymbol{U}_2, \boldsymbol{V}_2), \ldots, (\boldsymbol{U}_K, \boldsymbol{V}_K)\}$, where $\boldsymbol{U}$ is an image of size $H \times W \times C$, $\boldsymbol{V}$ is a tokenized text of length $L$. The goal is to learn a pair of encoders $f_\theta, g_\phi$, simultaneously: $\boldsymbol{U} \to \boldsymbol{u}, g_\phi : \boldsymbol{V} \to \boldsymbol{v}$ to map the image and text into an embedding space, $\boldsymbol{u}, \boldsymbol{v}$ are called embedding vectors of samples. A well-optimized visual-textual pre-trained model aligns the embedding vectors across the visual and textual models. That is, the embedding vectors extracted from semantically related image-text pairs earn higher similarity scores than the non-related ones. To generalize the problem, we view the embedding vectors as points on specified typologies. The similarity score between embedding vectors is an endowed distance function that evaluates the distance between the points. For instance, the commonly employed `cosine similarity` calculates the inner product of the normalized embedding vectors on the unit sphere as the (negative) distance between the sample pairs. To this end, we further consider the encoders as compositions of functions that i) map the inputs into the Euclidean space and ii) map the input vectors in Euclidean space on specified typologies. We denote this two-step mapping as: $f_\theta = \bar{f}_\theta \cdot f$, where $\bar{f}_\theta : \boldsymbol{U} \to \bar{\boldsymbol{u}}$ is the encoder with learnable parameters, $\bar{\boldsymbol{u}} \in \mathbb{R}^d$ is the output in the $d$-dimensional euclidean space. $f : \bar{\boldsymbol{u}} \to \boldsymbol{u}, \boldsymbol{u} \in \mathcal{M}$ is a specified operator (without learnable parameters) that maps the representations onto a topology $\mathcal{M}$, which is usually considered as a manifold embedded in the $d$-dimensional euclidean space.

**Contrastive Learning:** Following the definition in Oord et al. (2018); Wang & Isola (2020); Chen et al. (2021a); Radford et al. (2021), we formulate the contrastive loss as

$$\mathcal{L}_c(f_\theta, g_\phi; \tau, \mathcal{S}) := \mathbb{E}_{\substack{\boldsymbol{U},\boldsymbol{V} \sim \mathcal{S} \\ \boldsymbol{U}_i^- \neq \boldsymbol{U} \\ \boldsymbol{V}_j^- \neq \boldsymbol{V}}} \left[ -\log \frac{e^{-\tau d(f_\theta(\boldsymbol{U}), g_\phi(\boldsymbol{V}))}}{N} \right], \tag{1}$$

where $\tau$ is the temperature term, we write it as a multiplier for simplicity. $d(\cdot, \cdot)$ is the distance function between two points, and

$$N = \sum_{j \in [M]} e^{-\tau d(f_\theta(\boldsymbol{U}), g_\phi(\boldsymbol{V}_j^-))} + \sum_{i \in [M]} e^{-\tau d(f_\theta(\boldsymbol{U}_i^-), g_\phi(\boldsymbol{V}))},$$

is the negative term, with $M \in \mathbb{Z}^+$ denotes a fixed number of negative samples. Briefly, optimizing this loss term minimizes the distance between positive image-text pairs and maximizes the distance between negative image-text pairs. It is worth mentioning that, in recent studies Radford et al. (2021); Chen et al. (2021b), the contrastive loss is usually implemented as the cross-entropy between one-hot labels and the class probability obtained by `softmax` within a mini-batch $\mathcal{S}_M$. We also employ this implementation in this work as shown in Section 3.1, which can be formulated as

$$\mathcal{L}_c(f_\theta, g_\phi; \tau, \mathcal{S}) = \mathbb{E}_{\substack{\boldsymbol{U},\boldsymbol{V} \sim \mathcal{S}_M \\ i \in [M]}} \left[ H\left(\boldsymbol{q}_i | \sigma(\mathcal{U}_i)\right) + H\left(\boldsymbol{q}_i | \sigma(\mathcal{V}_i)\right) \right], \tag{2}$$

where $H(\cdot | \cdot)$ is the cross-entropy loss, $\mathcal{U}_i$, $\mathcal{V}_i$ are the (negative) distance between an $i$-th image/text to all the texts/images in the mini-batch. $\sigma$ is the `softmax` function, $\boldsymbol{q}_i$ is the one-hot label vectors of $i$.

**Oblique Manifold:** The properties of the oblique manifold for the machine learning community have been studied in past literature (Absil & Gallivan, 2006); it has a rich geometric structure allowing researchers to develop new algorithms for machine learning and other applications. In concise elucidation, the oblique manifold $\mathrm{Ob}(n, m)$ is a Riemannian manifold that assumes all its elements are matrices of size $n \times m$, while columns of these matrices possess a unitary norm. The oblique manifold can be viewed as a submanifold of a Euclidean space $\mathbb{R}^{n \times m}$, or a product manifold of spheres $\underbrace{\mathbb{S}^{n-1} \times \cdots \times \mathbb{S}^{n-1}}_{m \text{ copies}}$, where $\mathbb{S}^{n-1}$ is the sphere

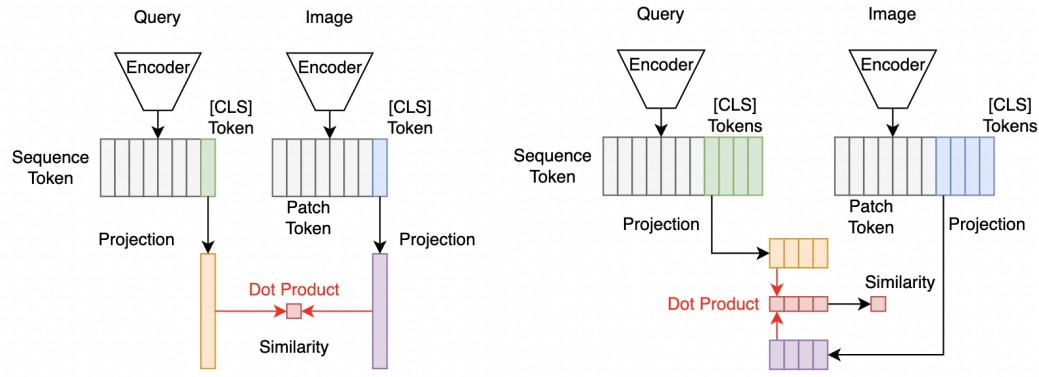

(a) Vanilla CLIP implementation        (b) Multi-token oblique implementation

Figure 1: Illustration of the system, the "Encoder" denotes the Bert and ViT models for textual (Query) and visual (Image) inputs, respectively. The tiles with colors denote the [CLS] token at the output layer and their projection for both modalities. The red arrow denotes the dot product between the vectors. In (b) multi-token oblique implementation, we sum up the dot products to obtain the similarity.

manifold embedded in $\mathbb{R}^n$. Therefore, we have a set of neat operations for the exponential and logarithmic mapping for the oblique manifold, which is the same as that of the sphere manifold, but are applied in a column-wise style. From an engineering perspective, this property allows us to project the feature vectors in Euclidean space onto the oblique manifold with a simple column-wise normalization operation, avoiding more complicated operations. Hence, Strictly, we follow the definition in Absil et al. (2009), that is,

$$\mathrm{Ob}(n, m) := \{\boldsymbol{X} \in \mathbb{R}^{n \times m} : \mathrm{diag}(\boldsymbol{X}^T \boldsymbol{X}) = \boldsymbol{I}_n\}, \tag{3}$$

where $\boldsymbol{I}_n$ is the identity matrix of size $n$. The diag$(\cdot)$ is the operator that spans a new diagonal matrix, with its diagonal elements identical to the original matrix, regardless of the values of other non-diagonal elements of $\boldsymbol{X}^T \boldsymbol{X}$.

**Transformer with [CLS] Token:** In the design of both the textual transformer (BERT, Kenton & Toutanova (2019)) and the visual transformer (ViT, Dosovitskiy et al. (2020)), *one* learnable embedding is used to represent global information, termed as [CLS] token. Different from the sequence (patch) tokens, the [CLS] token is a key component of the transformer encoder. It is randomly initialized and updated through gradient descent during the optimization. Furthermore, the [CLS] token holds a fixed position embedding, avoiding the influence of the positional information. Therefore, the [CLS] token is considered to participate in the computation of global attention. Finally, the state of the [CLS] token at the output layer is projected by an MLP to obtain feature representations.

## 3 Methodology

### 3.1 Proposed Approach for Implementation of the Oblique Topology

In this section, with the notations defined in Section 2, we describe our proposed approach for better contrastive alignment learning. Concisely, we employ the *oblique manifold* as the embedding space topology, with the (negative) *inner product* as the *distance* function. The oblique topology is implemented with two different approaches, using either multiple or single [CLS] tokens in the transformer encoders. We put an illustration of the vanilla CLIP system and the multiple [CLS] tokens oblique implementation in Figure 1. Their details are discussed below.

**Multi-Token Oblique Implementation:** In the vanilla CLIP implementation (Figure 1a), the [CLS] token at the output layer is firstly projected to a $l-$dimensional embedding space and then normalized; hence we have the feature representations encoded in the spherical space of $\mathbb{S}^{l-1}$ (embedded in $\mathbb{R}^l$). To construct the oblique topology, we extend the number of [CLS] tokens to the oblique $m$, that is, the copies of the sub-spheres in the oblique manifold. For the visual encoder, these [CLS] tokens are randomly initialized

```
# d      - dimension of the hidden embedding      # cls_U  - class tokens for image, [n, Ob_n, d]
# U      - mini-batch of image token, [n, p, d]   # cls_V  - class tokens for text, [n, Ob_n, d]
# V      - mini-batch of text token, [n, l, d]     # t      - learned temperature parameter
# Ob_m   - m of the oblique manifold (the dimension of each sub-sphere)
# Ob_n   - n of the oblique manifold (the number of additional tokens attached)

# concatenate cls_tokens and extract features
U_ = concatenate([cls_U, U], axis=1); u_bar = visual_transformer(U_) #[n, Ob_n + p, d]
V_ = concatenate([cls_V, V], axis=1); v_bar = textual_transformer(V_) #[n, Ob_n + l, d]

# map features onto Ob(n,m) and calculate distance
u = projection_u(u_bar[:Ob_n]).l2_normalize(axis=-1)  # [n, Ob_n, d] -> [n, Ob_n, Ob_m]
v = projection_v(v_bar[:Ob_n]).l2_normalize(axis=-1)  # [n, Ob_n, d] -> [n, Ob_n, Ob_m]
neg_distances = einsum('inm,jnm->ij', u, v) * t.exp() # [n, Ob_n, Ob_m], [n, Ob_n, Ob_m] -> [n, n]

# symmetric loss function
labels = arange(n) # 0, 1, ..., n-1
loss = (CE_loss(neg_distances, labels, axis=0) + CE_loss(neg_distances, labels, axis=1)) / 2
```

Figure 2: Python-like pseudo-code of the proposed approach.

to break symmetry, while for the textual encoder, we use different absolute positional embeddings for each [CLS] token. Given the fact that the dimension for the embedding space could be a critical factor to the performance of the system (Gu et al., 2021), we select the oblique $n$ with a conservative strategy. Specifically, we anchor the dimension of Euclidean spaces to be the same as the reference model, then vary the oblique $n, m$ such that $n \times m = l$. We denote this implementation as $\text{Multi}(n, m)$. The sub-spheres could benefit from the global attention operation and provide more representative feature embeddings. On the contrary, the multi-token implementation requires more computational resources in the backbone since the [CLS] tokens are involved in the computation of global attention.

**Single-Token Oblique Implementation:** In the Section 4.3 section, we employ a "clean" implementation to examine the properties of the embedding space topology, which brings no more parameters and keeps the same computational complexity. Concretely, we employ the original single [CLS] token CLIP system. To map the feature vectors onto the oblique manifold $\text{Ob}(n, m)$, we first reshape the feature vectors $\bar{\boldsymbol{u}}, \bar{\boldsymbol{v}}$ of size $d$ to a matrix of shape $m \times n$, then we $\ell_2-$normalize the columns.

**Distance Function:** The *distance* is defined as the negative inner product between two oblique manifolds. We compute it as the negative value of the trace of the matrix product, *i.e.* $d(\boldsymbol{u}, \boldsymbol{v}) = -\text{tr}(\boldsymbol{u}^T\boldsymbol{v})$. Here, $d : \text{Ob} \times \text{Ob} \to [-m, m]$ is a function that maps two oblique manifolds with size $n \times m$ into a real value. It is worth mentioning that, although such a definition is not a restricted metric or distance function for the oblique topology itself, it is still an available choice since we only require the symmetry property (last line in Section 3.1) of the mapping for the calculation of the cross-entropy loss function. That is, for any oblique manifolds $\boldsymbol{X}$ and $\boldsymbol{Y}$, $d(\boldsymbol{X}, \boldsymbol{Y}) = d(\boldsymbol{Y}, \boldsymbol{X})$. In Section 3.1, we employ the term "neg_distances" to avoid reduplicated calculation of the negative operation.

## 3.2   Rethinking The Properties of Topology

In this section, we discuss our motivation in detail. We unscramble three more neglected properties of the embedding space in addition to the distance range. To make the discussion clear, we compare the proposed approach with three reference configurations of different topologies and distance functions. Specifically, we consider i) the sphere $\mathbb{S}^{d-1}$ endowed with the inner product as distance, ii) the euclidean space $\mathbb{R}^d$ endowed with $\ell_2$ distance; iii) the oblique manifold $\text{Ob}(d/m, m)$ endowed with the minimizing geodesic as distance, which is denoted as $\text{Geo}(\boldsymbol{u}, \boldsymbol{v}) = \text{tr}^{\frac{1}{2}}(\arccos^2(\boldsymbol{u}^T\boldsymbol{v}))$. The comparison is summarized in Table 1. Keeping these properties favored for contrastive learning is important in the design of embedding topology.

**i. Low computational resource:** In the contrastive learning algorithm, the logits (or distance/similarity) matrix often costs the most computational resource in large-scale training. For instance, given a mini-batch of sample pairs of size $b$ with $d-$dimensional output, the computation of the inner product achieves a complexity of $O(b^2 d)$ and storage usage of $O(b^2)$. However, since the back-propagation of the $\ell_2-$norm requires intermediate results that cannot be "inplace" calculated, the $\ell_2-$norm (or any $\ell_p-$norm based distance) in Euclidean space requires a storage usage of $O(b^2 d)$. On the contrary, the geodesic distance cache $m$ curve lengths and hence requires $O(b^2 m)$ storage usage, while the proposed approach only needs one matrix multiplication after re-vectorization.

**ii. Proper definition of uniformity:** As explained by Wang & Isola (2020), contrastive loss is a combination of two objects a) alignment of features from positive sample pairs and b) the distribution of the features encouraged to match. Naturally, with the loss form defined in Equation (2), the distribution object will result in a uniform distribution on the sphere. Although it is not essential for the distribution of samples to be uniform as discovered by Chen et al. (2021a), it is necessary to define a proper prior distribution for samples to match via optimal transport algorithms (*e.g.* sliced Wasserstein distance), which is undoubtedly a computational burden. Both the sphere and oblique manifold have a proper uniform distribution defined as the surface area measure, while the unbounded Euclidean space does not. In practice, the $\ell_2-$norm defined distance between the samples grows larger along training and eventually overflows.

**iii. "Relaxed" triangular inequality:** Assume that we have a noisy dataset and a model $f_\theta^*, g_\phi^*$ that is "well-optimized" using this dataset. For this model, we have the following properties: For a positive pair $(U, V)^+$, their distance $d(u^*, v^*) = d(f_\theta^*(U), g_\phi^*(V))$ is upper-bounded by a small $\epsilon^+$, and the distance for negative pairs $(U, V)^-$ is lower-bounded by a large $\epsilon^-$. Now, let us consider the following scenario, accidentally, in a set of two pairs of its training samples $\mathcal{S}_\pm = \{(U_1, V_1), (U_2, V_2)\}$, the pair $(U_1, V_2)^-$ is also semantically correlated, *despite of being recognized as a negative sample*. This "well-optimized" model will predict a distance upper-bounded by $\epsilon^+$ for this pair instead of a larger value than $\epsilon^-$. If the distance function $d$ is a *metric*, then according to the triangle inequality axiom of metric, we have the following inequality (see Figure 3 for an intuitive illustration),

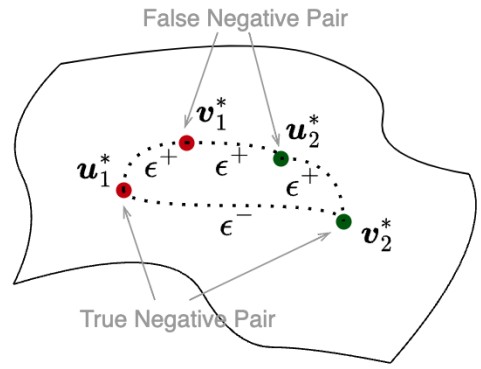

Figure 3: Illustration of the bounded negative distance between true negative pair of samples.

$$\epsilon^- \le d(u_2^*, v_1^*) \le d(u_1^*, v_2^*) + d(u_2^*, v_2^*) + d(u_1^*, v_1^*) \le 3\epsilon^+ \quad (4)$$

From this simple derivation, we find that the negative distance is bounded by three times the positive distance, once false negative pairs of samples are involved. Given the fact that, in a mini-batch of sufficiently large size, it is pervasive to have negative pairs of image and text that match each other equally well as the positive ones, in the noisy training dataset (even in these human-labeled datasets, see Chun et al. (2022)). As a consequence, such a "well-optimized" model would not learn numerical separated distance ranges for positive and negative pairs to minimize the contrastive loss (See Figure 6). Therefore, we need to "relax" the triangular inequality to alleviate the effects of ambiguity of the positive/negative pairs (if we do not scale the distance range using the temperature). In practice, we have the following observations: i) The positive pairs of samples usually have a much larger distance than that in the perfect alignment condition, ii) $d(u_1^*, u_2^*)$ may become exceptionally small since none of the loss terms regularize it, resulting in a further tightened bound of the negative distance; iii) Although the inner product distance is not a *metric*, it still obeys an "relaxed" triangular inequality because we can yield a metric on the sphere by `ArcCos` (the geodesic), see Schubert (2021) for more information. In Table 1, we characterize both the minimizing geodesic and the $\ell_2-$norm distances are metrics and hence obey the restricted triangular inequality.

**iv. Broad distance range:** As discussed in Section 1, a broad distance range helps the system achieve equilibrium under noisy samples. In the experimental analysis, we show that the oblique embedding space could learn the proper temperature much faster than the spherical embedding space because of the magni-

| Topology | Sphere $\mathbb{S}^{d-1}$ | Euclidean $\mathbb{R}^d$ | Oblique$(d/m, m)$ | Oblique$(d/m, m)$ |
|---|---|---|---|---|
| Distance | $-\boldsymbol{u}^T\boldsymbol{v}$ | $\|\boldsymbol{u} - \boldsymbol{v}\|_2$ | $\mathrm{Geo}(\boldsymbol{u}, \boldsymbol{v})$ | $-\mathrm{tr}(\boldsymbol{u}^T\boldsymbol{v})$ |
| Memory Resource | $O(b^2)$ | $O(b^2 d)$ | $O(b^2 m)$ | $O(b^2)$ |
| Uniformity | surface measure | undefined | surface measure | surface measure |
| Inequality | relaxed | restricted | restricted | relaxed |
| Distance Range | $[-1, 1]$ | $[0, +\infty)$ | $[0, m\pi]$ | $[-m, m]$ |

Table 1: Summary of different topologies endowed with different distances. The total dimension of the embedding vector is denoted as $d$. The mini-batch size is denoted as $b$. Memory Resource denotes the cost of the similarity matrix. Green box stands for the properties that are *favored* for contrastive learning. Red box stands for the properties that are *unfavored* for contrastive learning. Best viewed in color.

fied distance range ($[-m, m]$). More interestingly, we show that the unbounded distance range allows the Euclidean space learns equally well without using a temperature parameter.

## 4 Experimental Analysis

### 4.1 Experimental Settings

**Datasets:** For the experimental analysis in Section 4.1, we collect data from publicly available datasets Schuhmann et al. (2021); Changpinyo et al. (2021); Sharma et al. (2018); Chen et al. (2015); Krishna et al. (2017); Plummer et al. (2015); Chen et al. (2015); Russakovsky et al. (2015); Desai et al. (2021); Kuznetsova et al. (2020); Li et al. (2017)[1]. We also have clawed weakly related image-text pairs from the web, resulting in a total of 420 million individual images and roughly 500 million image-text pairs. This dataset is comparable to the one employed in the official CLIP paper Radford et al. (2021) and another open source re-implementation Ilharco et al. (2021). To further remove the bias caused by datasets, we also re-implement the naive clip algorithm for reference. We evaluate the proposed methods with two types of vision tasks: i) `Zero-Shot` image-to-text and text-to-image retrieval on Flickr30k (Plummer et al., 2015) and MSCOCO Lin et al. (2014) ii) `Zero-Shot` classification on ImageNet-1K (Russakovsky et al., 2015), ImageNet-V2 Recht et al. (2019), ImageNet-R Hendrycks et al. (2021a) and ImageNet-A Hendrycks et al. (2021b).

For the experimental analysis in Section 4.3, we employ the 15M subset (Cui et al., 2022) of the YFCC100M dataset (Thomee et al., 2016) as the training dataset, which contains roughly 15.3 million internet collected weakly related image-text pairs. Furthermore, we employ the RedCaps (Desai et al., 2021) dataset as the out-domain data for visualizing the distributions of sample distances. In addition to Zero-Shot evaluation, we also provide `Linear Probe` performance for reference.

**Models:** Due to the limited computational resources, we adopt a moderate scaling of the models. Specifically, For the ablation experiments, we employ the original ViT-S/16 architecture for our image encoders Dosovitskiy et al. (2020), with an input image resolution of 224, resulting in 196 image tokens. For large-scale training, we employ the ViT-B/16 as our image encoders. For our text encoders, we employ Ernie-2.0-en-base (Sun et al., 2020), which is literally a Bert model (Devlin et al., 2018) of 12 layers and 512 hidden neuron sizes with a customized vocabulary of 30,522 tokens, and the maximum context length is set to be 77. We project the feature representation ([CLS] token) from the top layer of transformers to a (sum of) 512-dimensional embedding space. All the parameters except the temperature are optimized from random initialization. The default initialization of the project matrix employs the Gaussian initializer of zero mean, and standard deviation equal reversed square root of the input size (*a.k.a.* Kaiming initialization).

---

[1]The availability of LAION400M is about 90%, so we decided to use some collectable public datasets.

| Method *baseline[impl.]* | IN ZS cls. Acc@1 | INV2 ZS cls. Acc@1 | IN-A ZS cls. Acc@1 | IN-R ZS cls. Acc@1 | Flickr30K Zero-shot | | | MSCOCO* Zero-shot | | |
|---|---|---|---|---|---|---|---|---|---|---|
| | | | | | I2T R@1 | T2I R@1 | Mean R@1/5/10 | I2T R@1 | T2I R@1 | Mean R@1/5/10 |
| *ViT-B/16-224 as visual bone.* | | | | | | | | | | |
| CLIP[openAI[†]] | 68.7 | 61.9 | 50.1 | 77.7 | 81.9 | 62.1 | 86.1 | 55.4 | 38.4 | 66.3 |
| CLIP[openCLIP[‡]] | 67.0 | 59.6 | 33.2 | **77.9** | 83.2 | 65.5 | 87.6 | 52.4 | 38.4 | 62.4 |
| CLIP[our-impl.] | 69.5 | 61.4 | 49.5 | 70.6 | 84.2 | 61.7 | 86.4 | **64.1** | **43.9** | **72.4** |
| CLIP[Multi(32,16)] | **76.4** | **68.0** | **55.8** | 75.2 | **85.2** | **66.3** | **88.3** | 63.8 | 42.9 | **72.4** |
| *ViT-L/14-224 as visual bone for reference.* | | | | | | | | | | |
| CLIP[openAI[†]] | 75.5 | 69.7 | 70.7 | 87.9 | 85.0 | 65.2 | 87.7 | 56.3 | 36.5 | 65.2 |
| CLIP[openCLIP[‡]] | 72.7 | 65.6 | 46.6 | 84.8 | 87.6 | 70.3 | 90.1 | 59.7 | 43.0 | 70.0 |

Table 2: Comparsion of large scale contrastive visual-textual pre-train model on benchmark datasets. [†] and [‡] denote the implementation from Radford et al. (2021) and Ilharco et al. (2021), respectively. The metric **Mean** stands for the average value of R@1/5/10 of I2T/T2I retrieval performance. * denotes the Karpathy test split Karpathy & Fei-Fei (2015).

For the temperature, we initialize it with $e^t$ for $t = 0.0, 2.64, 5.31$. Hyper-parameters employed for training are provided in the appendix. The details of the hyperparameters are provided in Table 5 in the appendix.

**Evaluation:** For zero-shot retrieval on Flickr30K and MSCOCO, we employ the logits (distance) computed by the distance function and report the image-text pairs with the top-$k$ shortest distance as the retrieval results. For zero-shot classification on ImageNet. We employ multiple prompt templates described in Radford et al. (2021), while we first compute the distances between image and text embeddings, then average the distances. For linear probe classification on ImageNet, we remove the learned projection head (no topological structure is preserved), then attach a random initialized linear projector to map the feature representation to the 1,000 class logits.

## 4.2 Main Results using Large-Scale Dataset

We compare the performance of the proposed method using the larger scale configuration, which matches the publicly released ones in teams of datasets samples, model sizes, and training progress. We evaluate the learned models on the commonly employed image classification and image-text retrieval tasks. The results are reported in Table 2, with the ViT-B/16 as the visual backbone; we re-implement the naive CLIP model as the reference, which holds a similar performance as the publicly released ones. On the other hand, our proposed model with the multi-token implementation of (32,16) significantly outperforms the other ViT-B/16 models in general, with less than 8% more computational costs. The only exception is the top-1 retrieval performance on the MSCOCO datasets. The reason could be two-folded. Firstly, we observe a mild "semantic decoupling" between the embedding of tokens through the visualization (see Section 4.5), that is, some of the individual [CLS] tokens focus on specified objects and provide a high alignment confidence. This may cause confusion in understanding the given scene as a whole; hence the recall@top-1 performance is degraded. Secondly, the most suitable temperature during training for aligning object-level and scene-level concepts might differ. In our experiment, we decrease the upper limitation of the temperature to 6.25 (100 / 16 [tokens]) since the oblique topology owns the border distance range. The scene-level concept alignment might require a larger temperature for "ambiguity" to achieve better retrieval performance.

## 4.3 Ablational Experiments on Properties of Geometry

In this section, we conduct a series of experiments, to examine how the mentioned properties in Table 1 affect the performance of the contrastive alignment. We employ the oblique manifold structure of $n = 64$, $m = 8$ for our proposed approach, denoted as Ob(64,8). Since this is a single-token implementation, it won't benefit from the extra computational complexity, and the only difference is the shape of the embedding space topology. All the results are presented within Table 3, and subsequent sections will elucidate the reader on how to interpret this tabular data.

| Topology | Distance | Zero-Shot I2T R@1 | Zero-Shot T2I R@1 | Zero-Shot Cls. Acc. | Linear Probe Cls. Acc. |
|---|---|---|---|---|---|
| | | Temperature, init=$e^{2.64}$, gradient=True | | | |
| Sphere | $-\boldsymbol{u}^T\boldsymbol{v}$ | 48.3 | 31.45 | 30.62 | 60.38 |
| Euc | $\|\boldsymbol{u}-\boldsymbol{v}\|_2$ | 47.9 | 32.29 | 30.36 | 59.90 |
| Ob(64, 8) | $\mathrm{Geo}(\boldsymbol{u},\boldsymbol{v})$ | 50.7 | 32.35 | 30.79 | 60.43 |
| Ob(64, 8) | $-\mathrm{tr}(\boldsymbol{u}^T\boldsymbol{v})$ | 52.3 | 32.89 | 30.70 | 60.32 |
| | | Temperature, init=$e^{5.31}$, gradient=True | | | |
| Sphere | $-\boldsymbol{u}^T\boldsymbol{v}$ | 46.8 | 31.13 | 29.60 | 59.82 |
| Euc | $\|\boldsymbol{u}-\boldsymbol{v}\|_2$ | 48.5 | 32.69 | 30.68 | 59.74 |
| Ob(64, 8) | $\mathrm{Geo}(\boldsymbol{u},\boldsymbol{v})$ | 50.7 | 31.59 | 30.34 | 59.60 |
| Ob(64, 8) | $-\mathrm{tr}(\boldsymbol{u}^T\boldsymbol{v})$ | 50.3 | 33.37 | 30.23 | 59.76 |
| | | Temperature, init=$e^{0.0}$, gradient=True | | | |
| Sphere | $-\boldsymbol{u}^T\boldsymbol{v}$ | 49.0 | 30.33 | 28.59 | 59.56 |
| Euc | $\|\boldsymbol{u}-\boldsymbol{v}\|_2$ | 47.4 | 30.71 | 29.85 | 60.09 |
| Ob(64, 8) | $\mathrm{Geo}(\boldsymbol{u},\boldsymbol{v})$ | 49.9 | 32.49 | 30.21 | 60.61 |
| Ob(64, 8) | $-\mathrm{tr}(\boldsymbol{u}^T\boldsymbol{v})$ | 50.9 | 32.71 | 30.50 | 60.66 |
| | | Temperature, init=$e^{0.0}$, gradient=False | | | |
| Sphere | $-\boldsymbol{u}^T\boldsymbol{v}$ | 5.1 | 3.461 | 4.04 | 45.37 |
| Euc | $\|\boldsymbol{u}-\boldsymbol{v}\|_2$ | 47.6 | 30.43 | 29.51 | 59.20 |
| Ob(64, 8) | $\mathrm{Geo}(\boldsymbol{u},\boldsymbol{v})$ | 4.1 | 2.921 | 3.10 | 21.67 |
| Ob(64, 8) | $-\mathrm{tr}(\boldsymbol{u}^T\boldsymbol{v})$ | 30.3 | 18.48 | 21.20 | 57.93 |

Table 3: The retrieval and classification performance of different configurations under different temperature initialization conditions. "gradient={True/False}" donates if the temperature is learnable.

**Effects of the Uniformity:** To examine the effects of uniformity, we compare the performance between the Euclidean with $\ell_2$ distance and the Oblique with geodesic distance. These configurations own restricted triangular inequality and border distance range in common; the only difference is that uniformity can be defined on the Oblique. In general, when the temperature is learnable and initialized with a decent value, the performance of the Oblique with geodesic configuration is better than the Euclidean with $\ell_2$ configuration. These results indicate the importance of properly defined uniformity.

**Effects of the Tri-angular Inequality:** Next, we examine the effects of the triangular inequality using the Oblique topologies endowed with different distance functions, that is, the geodesic distance and the inner product distance. The results are visually depicted in Table 3, specifically between the 3rd and 4th lines of each data block presented in the table. Upon observation, it becomes evident that the inner product distance outperforms the geodesic distance on average in both retrieval and linear probe tasks. Moreover, when the temperature is unlearnable (the last block), the inner product distance still provides the model trainability, showing the advantage of removing the restriction of tri-angular inequality.

**Effects of the Distance Range:** Finally, we present the effects of the distance range by comparing the proposed oblique topology with inner product distance and the baseline spherical topology. Notably, under all the temperature initialization schemes, our proposed methodology demonstrates a commendable enhancement in the top-1 recall accuracy of +4.0%/1.44%, +3.5%/2.24%, and +1.9%/2.38%, in contrast to the baseline approach for the retrieval tasks. This suggests that a larger distance range helps the alignment of the features from different modalities. There is one more piece of evidence that lies in the last block, the Euclidean with $\ell_2$ distance configuration obtains consistent performance regardless of the temperature parameter, as it operates with an unrestricted distance range.

### 4.4 Miscellaneous Experiments on System Design

**Effects of temperature initialization:** In Table 6, we re-organize the results in Table 3, to demonstrate of the effects of temperature initialization on certain configurations. Although it can be seen that the final

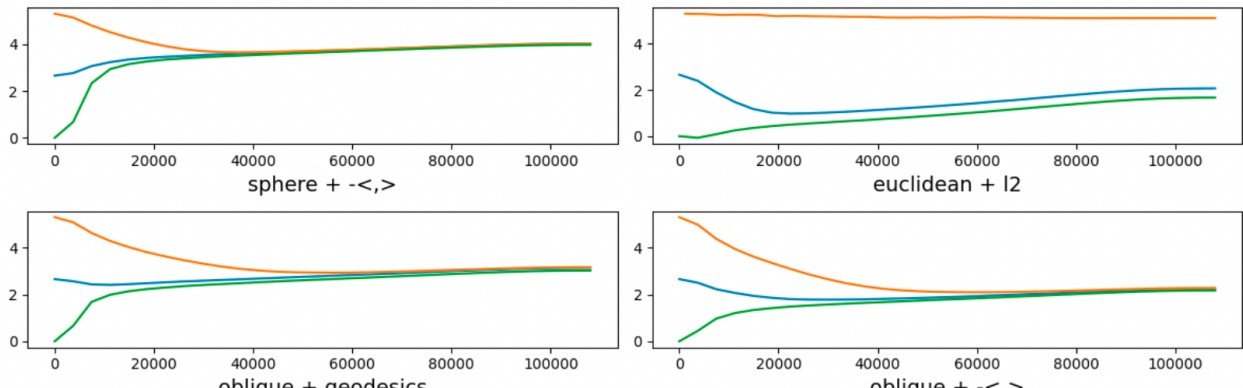

Figure 4: The learning curve of the temperature. -<,>, l2 and geodesics denote the negative inner product, $\ell_2$, and minimizing geodesic distance, respectively. The orange, blue, and green curves denote the initialization of $e^5.31$, $e^2.64$, and $e^0$, respectively.

| Topology | Distance | Zero-Shot I2T R@1 | Zero-Shot T2I R@1 | Zero-Shot Cls. Acc. | Linear Probe Cls. Acc. |
|---|---|---|---|---|---|
| \multicolumn{6}{c}{Temperature, init=$e^{2.64}$, gradient=True} | | | | | |
| Sphere(512) | $-\boldsymbol{u}^T\boldsymbol{v}$ | 48.3 | 31.45 | 30.62 | 60.38 |
| Ob(256, 2) | $-\text{tr}(\boldsymbol{u}^T\boldsymbol{v})$ | 48.0 | 32.25 | 30.33 | 60.52 |
| Ob(64, 8) | $-\text{tr}(\boldsymbol{u}^T\boldsymbol{v})$ | **52.3** | 32.89 | 30.70 | 60.32 |
| Ob(16, 32) | $-\text{tr}(\boldsymbol{u}^T\boldsymbol{v})$ | 50.4 | **33.01** | **30.93** | **60.79** |
| Ob(4, 128) | $-\text{tr}(\boldsymbol{u}^T\boldsymbol{v})$ | 48.2 | 32.91 | 30.57 | 59.99 |
| Multi(256, 2) | $-\text{tr}(\boldsymbol{u}^T\boldsymbol{v})$ | 49.2 | 32.29 | 30.04 | 61.59 |
| Multi(64, 8) | $-\text{tr}(\boldsymbol{u}^T\boldsymbol{v})$ | 54.0 | **34.27** | **31.93** | 62.41 |
| Multi(16, 32) | $-\text{tr}(\boldsymbol{u}^T\boldsymbol{v})$ | **54.0** | 33.43 | 30.88 | **63.71** |

Table 4: The retrieval and classification performance of the proposed approach using different oblique manifold structures and the multi-token implementation. "gradient={True/False}" donates if the temperature is learnable.

performance is not largely impacted by the initialization, a large or small temperature at the start of the training still cause difficulties in optimizing. It is also interesting to see that when the temperature is fixed at 1.0, the sphere one (narrow distance range) and the geodesic one (restrict triangular inequality) fail to learn meaningful feature embeddings for contrastive alignment. The performance of the proposed approach is also largely reduced. However, since the Euclidean topology does not have an upper bound of the distance, the optimizer can still reach the equilibrium. We further draw the trend of the temperature during the training progress in Figure 4. From the figures, we can confirm that: i) Given a bounded distance range, the temperature is an inherent property of the datasets, depicting the noise level of the datasets; ii) The temperature will first converge to an equilibrium regardless of initialization, then raise gradually along the optimization progress; iii) It takes longer training iterations for the temperature to converge on the oblique embedding space, while the final temperature is smaller than that in the sphere embedding space. The results suggest that border distance range and topology structure help the model focus more on aligning images and texts rather than finding the equilibrium.

**Ablation on oblique structures and multi-token implementation:** In Table 4, we modify the structure of the oblique manifold under fixed total dimensions. It can be seen that a higher $m$ value (*i.e.* the number of product sub-spheres) is more likely to obtain a better zero-shot classification accuracy and text-to-image retrieval recall. We, therefore, conjecture that the broader distance range helps the system reach equilibrium faster. However, an over-complicated structure such as Ob(4,128) could ruin the performance. The possible

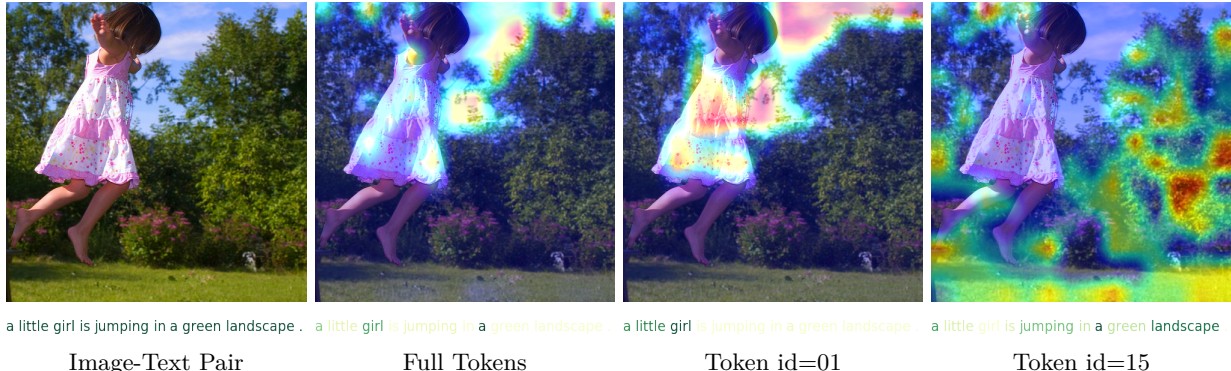

a little girl is jumping in a green landscape .   a little girl is jumping in a green landscape .   a little girl is jumping in a green landscape .   a little girl is jumping in a green landscape

| Image-Text Pair | Full Tokens | Token id=01 | Token id=15 |

Figure 5: Visualization of the importance map using the Grad-CAM algorithm. The columns from left to right stand for: the input image-text pair; the importance map computed based on the final matching score; the importance maps based on the matching scores of two individual tokens and the involved token ids. Additational results are provided in Appendix A.8

reason is that each sub-sphere $\mathbb{S}^{d-1}$ that is embedded in $\mathbb{R}^d$ has one less effective dimension. Therefore, the oblique structure with large numbers of sub-spheres may perform worse.

We also provide the ablation results of different multi-tokens oblique structure implementations in the table's lower half, denoted as Multi$(\cdot, \cdot)$. We concatenate all the representations together for the linear probe before projecting them to 1,000 class logit. It can be seen that the multi-token oblique implementations consistently outperform their single-token versions. Notably, since the increased number of parameters for the class tokens $(n \times d)$ is negligible compared to that of the overall system, we consider the participants of class tokens in global attention as the primary reason for the performance boost.

### 4.5 Visualization on Tokens Attention Regions

In this section, we provide a commonly adopted neural network explanation method to visualize the influence of inputs on the final outcome. Specifically, we employ the Grad-CAM Selvaraju et al. (2017) algorithm to highlight the interested parts by the model of both images and their corresponding texts. Notably, the original design of the Grad-CAM algorithm precludes its direct application to textual data. Therefore, we enhance its capabilities such that it also highlights the contributing parts of texts in a token-wise style. We employ examples from the evaluation set of the Flickr30K and MSCOCO datasets for visualization. The results are shown in Figure 5. Through our observations, we have noted that certain pairs of [CLS] tokens exhibit independent alignment, thereby reflecting a distinct concept or idea embedded within the image. We attribute the improved classification performance to this phenomenon, as it enables a more effective representation and understanding of the underlying content by leveraging the distinctive alignment of the [CLS] token pairs. It is noteworthy that the intrinsic decoupling phenomenon observed within the embeddings of [CLS] tokens is *NOT* universally present across all image-text pairs or within every token of an image-text pair. This is due to the inherent challenges faced by visualization algorithms in achieving precise correspondence in the importance maps for intricate semantic representations.

## 5 Related Works

**Momentum distillation:** In recent works such as Cheng et al. (2021); Li et al. (2021a), the momentum (self-)distillation is introduced to mitigate the semantic noise in the sample pairs. That is, a momentum version of the model is updated by the moving average of the model's historical parameters. Then, the cross entropy between the softmax logits computed by the model and its momentum version is used as an additional loss for supervision. The authors claim that the pseudo-targets of the momentum (self-)distillation will not penalize the model for matching negative samples that are reasonably similar. Here, we consider that the pseudo-targets do "relax" the triangular inequality restriction implicitly by letting the distance of

alignment be reasonably large. Hence, it could be much easier for the optimizer to find the equilibrium discussed in Section 3.2.

**Other implementation of non-metric distance:** In Yao et al. (2021), the authors proposed a so-called fine-grained contrastive learning scheme that matches all the visual and textual tokens using a maximum-average operator. Concretely, for each visual token, it finds the textual token with maximum similarity, then takes the average over the visual tokens as the similarity of the image to a text and vice versa. Using our framework, this work can be explained as embedding samples onto the product manifold $\mathbb{S}^{d-1} \times \cdots \times \mathbb{S}^{d-1}$ endowed with the maximum-average distance, which is a non-metric distance. At the same time, the authors employ the sub-manifold $\mathbb{S}^{d-1}$ to represent local information.

**The effects of softmax temperature:** In Wang & Liu (2021), the authors draw the uniformity of the embedding distribution and the tolerance to semantically similar samples of learned models under different temperatures. From the observations, the authors claim that "a good choice of temperature can compromise these two properties properly to both learn separable features and tolerant to semantically similar samples, improving the feature qualities and the downstream performances". Unlike our work, this work is done under uni-modal contrastive learning, where the semantic correlation of the negative samples is not a property of the datasets but rather a drawback of the larger mini-batch size.

**Uni-modal side tasks:** In works such as Mu et al. (2021); Li et al. (2021b); Yang et al. (2022), authors combine cross-modal contrastive loss with other uni-modal tasks, for instance, visual/textual self-supervised contrastive learning, masked image/language modeling. These combined methods empirically demonstrate superior performance in downstream tasks such as zero-shot classification. Although these works do not overlap with this one, we find that the uni-modal tasks provide reasonable uniformity within the visual/textual feature embedding, contrary to the cross-modal contrastive shown in Section 3.2. Therefore, the model could obtain a more "numerically relaxed" triangular inequality when dealing with noisy pairs of samples.

**Other works that employ oblique manifold:** It is notable that learning representations embedded on the oblique manifold for computer vision tasks have been explored by former studies. For instance, in Qi et al. (2021), the authors employ the oblique topology for few-shot learning. However, different from these works, our paper mainly tackles the noisy database problem in the contrastive image-text alignment task. We employ the oblique topology with a non-metric distance function to tackle the out-of-range equilibrium problem.

**Hyperbolic embedding space:** The hyperbolic topology is another popular choice for hierarchical representation embedding space, in both NLP (Nickel & Kiela, 2017), CV (Zhang et al., 2022; Liu et al., 2020; Khrulkov et al., 2020), and cross-modal (Guo et al., 2021) tasks. However, the hyperbolic topology has unfavored properties similar to Euclidean space. Its resource required for computing distance is high; it is difficult to define/implement uniformity in terms of numerical stability; also, the geodesic distance has restricted triangular inequality. Therefore, we do not consider this topology in this study.

## 6 Conclusion

**Summary:** This work discusses the essential properties of the feature embedding space for contrastive alignment. We show that the most commonly adopted cosine similarity has disadvantages in dealing with noisy data and training stability. Therefore, we propose to combine the oblique manifold with the negative inner product distance to tackle these problems. We employ multiple class tokens to implement the approach, which performs better in various zero-shot classification and image-text retrieval tasks practically.

**Limitation:** First, due to remarkably limited computational resources (and time), we cannot conduct experiments on a larger scale regarding batch size, training data, and parameters in the neural network. Second, in recent studies, besides the contrastive alignment, more pre-training tasks are appended to the head of the model using the non-normalized full token embedding. Such as image-text matching (Li et al., 2021a; Yang et al., 2022), image captioning (Yu et al., 2022), or masked modeling that do not employ the contrastive alignment (Wang et al., 2022). The performance improvement resulting from a better contrastive alignment could be marginal in these configurations. And hence leave future work on designing the topology of the full token embedding.

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

# A  Appendix

## A.1  Detailed Training Hyper-parameters Used in Section 4

| Hyperparameters | Value for Naive CLIP | Value for CLIP-Multi(32,16) | Value for Ablation Table 3 | Value for Ablation Table 4 |
|---|---|---|---|---|
| Batch size | 32,768 | 32,768 | 2,048 | 2,048 |
| Vocabulary size | 30,522 | 30,522 | 30,522 | 30,522 |
| Training epochs | 32 | 32 | 15 | 15 |
| Number [CLS] Tokens | 1 | 16 | 1/8 | 2/8/32/128 |
| Projection dims | 512 | 32 | 512/64 | 256/64/16/4 |
| Maximum temperature | 100.0 | 3.95 | 100.0 | 100.0 |
| Weight decay | 0.2 | 0.2 | 0.5 | 0.5 |
| Warm-up iterations | 2,000 | 2,000 | 5,000 | 5,000 |
| Peak Learning Rate | 0.0005 | 0.0005 | 0.0005 | 0.0005 |
| Adam $\beta_1$ | 0.9 | 0.9 | 0.9 | 0.9 |
| Adam $\beta_2$ | 0.998 | 0.998 | 0.98 | 0.98 |
| Adam $\epsilon$ | $10^{-8}$ | $10^{-8}$ | $10^{-8}$ | $10^{-8}$ |
| Gradient global norm | 1.0 | 1.0 | 1.0 | 1.0 |
| GPUs | 128×Nvidia-A100 | 128×Nvidia-A100 | 32×Nvidia-V100 | 32×Nvidia-V100 |
| Train Time | ∼5 days | ∼5 days | ∼1 day | ∼1 day |

Table 5: Detailed hyper-parameters used for in the experimental analysis.

We provide the hyper-parameters employed in the experiments in Table 5. We follow most of the hyper-parameters employed in the original CLIP (Radford et al., 2021) paper for both Naive CLIP re-implementation and our multi-token and single-token oblique implementation. We provide the details of the hyper-parameters for large-scale and ablation experiments below.

**Large-scale Experiments:** We train with a batch size of 32,768 and the AdamW optimizer (Loshchilov & Hutter, 2017) in all the large-scale experiments. We apply the standard training scheme of the original CLIP model, which contains 32 epochs of training. We did not employ mixed precision to reduce the possible overflow introduced by randomness for a stable reproduction. We set the $\beta_1 = 0.9$, $\beta_2 = 0.998$, $\epsilon = 1e$-8 in AdamW, and weight decay $= 0.2$ to further improve the stability. We use the cosine learning rate decay scheme of peak learning rate equal to 5$e$-4, combined with a warmup period of 2,000 iterations. For data augmentation, we only apply the `RandomResizedCrop` with a scale range of $[0.8, 1.0]$. Finally, in our multi-token oblique implementation, we reduce the maximum temperature to 3.95 due to its border distance range. This is a value obtained from the ablation study from Appendix A.4.

**Ablation Experiments:** We train with a batch size of 2,048 and the AdamW optimizer (Loshchilov & Hutter, 2017) in all the ablation experiments. We apply a compact training scheme that updates the model for 108,000 iterations, which is roughly equal to training the model for 15 epochs of the dataset. Since this is a fast training scheme, we set the $\beta_1 = 0.9$, $\beta_2 = 0.98$, $\epsilon = 1e$-8 in AdamW, and weight decay $= 0.5$, such that the training could converge faster in a stable approach. We use the cosine learning rate decay scheme of peak learning rate equal to 5$e$-4, combined with a warmup period of 5,000 iterations. In the linear probe evaluation, the hyperparameters follow the setup of MoCo v3 (Chen et al., 2021b). Concretely, we use SGD without momentum and no weight decay. The learning rate is schemed by cosine decay with a peak learning rate equal to 1.0, combined with a warmup period of 5 epochs. We train for 100 epochs and augment the image using the `RandomResizedCrop` with a scale range of $[0.75, 1.0]$ and `AutoAugment` with the code `rand-m9-mstd0.5-inc1`. In the ablation experiments, we do not change the maximum temperature clip value, leaving it the same for all topology configurations.

| Temp. Init. | Temp. Final | Converge Step | Zero-Shot I2T R@1 | Zero-Shot T2I R@1 | Zero-Shot Cls. Acc. | Linear Cls. Acc. |
|---|---|---|---|---|---|---|
| | | Topology: Sphere, | | Distance: $-\boldsymbol{u}^T\boldsymbol{v}$ | | |
| 2.659 | 4.033 | 18k | 48.3 | 31.45 | 30.62 | 60.38 |
| 5.310 | 4.021 | 39k | 46.8 | 31.13 | 29.60 | 59.82 |
| 1.000 | 3.976 | 22k | 49.0 | 30.33 | 28.59 | 59.56 |
| 1.000 | 1.000 | Detach | 5.1 | 3.461 | 4.04 | 45.37 |
| | | Topology: Euclidean, | | Distance: $\|\boldsymbol{u}-\boldsymbol{v}\|_2$ | | |
| 2.659 | 2.067 | 21k | 47.9 | 32.29 | 30.36 | 59.90 |
| 5.310 | 5.107 | 1k | 48.5 | 32.69 | 30.68 | 59.74 |
| 1.000 | 1.668 | 25k | 47.4 | 30.71 | 29.85 | 60.09 |
| 1.000 | 1.000 | Detach | 47.6 | 30.43 | 29.51 | 59.20 |
| | | Topology: Ob(64, 8), | | Distance: $\mathrm{Geo}(\boldsymbol{u},\boldsymbol{v})$ | | |
| 2.659 | 3.135 | 20k | 50.7 | 32.35 | **30.79** | 60.43 |
| 5.310 | 3.168 | 55k | 50.7 | 31.59 | 30.34 | 59.60 |
| 1.000 | 3.024 | 42k | 49.9 | 32.49 | 30.21 | 60.61 |
| 1.000 | 1.000 | Detach | 4.1 | 2.921 | 3.10 | 21.67 |
| | | Topology: Ob(64, 8), | | Distance: $-\mathrm{tr}(\boldsymbol{u}^T\boldsymbol{v})$ | | |
| 2.659 | 2.231 | 24k | **52.3** | 32.89 | 30.70 | 60.32 |
| 5.310 | 2.280 | 57k | 50.3 | **33.37** | 30.23 | 59.76 |
| 1.000 | 2.174 | 36k | 50.9 | 32.71 | 30.50 | **60.66** |
| 1.000 | 1.000 | Detach | 30.3 | 18.48 | 21.20 | 57.93 |

Table 6: The retrieval and classification performance of different configurations under different temperature initialization conditions. The performance report in this table is the same as Table 3, but is aggregated by topologies. "Temp. Init." denotes the values for initializing temperature; "Temp. Final" denotes the final temperature at the end of training; "Converge Step" denotes the number of steps for temperature starts to converge (changes less than 2% for an epoch.)

## A.2    Table 2 from the View of Topologies

In Table 6, we review Table 3 by the topologies. We further provide the final temperature at the end of training and at what step the temperature converges (changes less than 2% for an epoch, also see Figure 4). It can be seen that the performance of the Euclidean topology is only slightly affected by the initialization of the temperatures, and even though the temperature is detached from learning, it still performs reasonably well because of the unlimited distance range. At the same time, the spherical and oblique topologies are affected by how the temperature is initialized. However, a rough trend can be seen that the faster the temperature converges, the better performance the model achieves, which means the learnable temperature delays the learning of the methods. The model needs first to find a proper temperature and then begin to learn representations well.

## A.3    Distribution of Learned Distance

We depict the distribution of distance for pairs of samples in Figure 6. As argued in Section 3.2 of the main manuscripts, since the cross-modal contrastive loss does not handle the uni-modal data distributions, the distance between negative pairs of images and texts could be much smaller than that of a positive image-text pair, resulting in a tighter distance bound. Also, we can see this phenomenon is much more severe in out-domain data, which could reduce the transferability of the feature embeddings to downstream tasks. It is also notable that, the oblique endowed with the negative inner product as the distance function learns similar distributions compared to the sphere reference, while the numerical values of distances between samples are inherently larger without having multiplied with temperature.

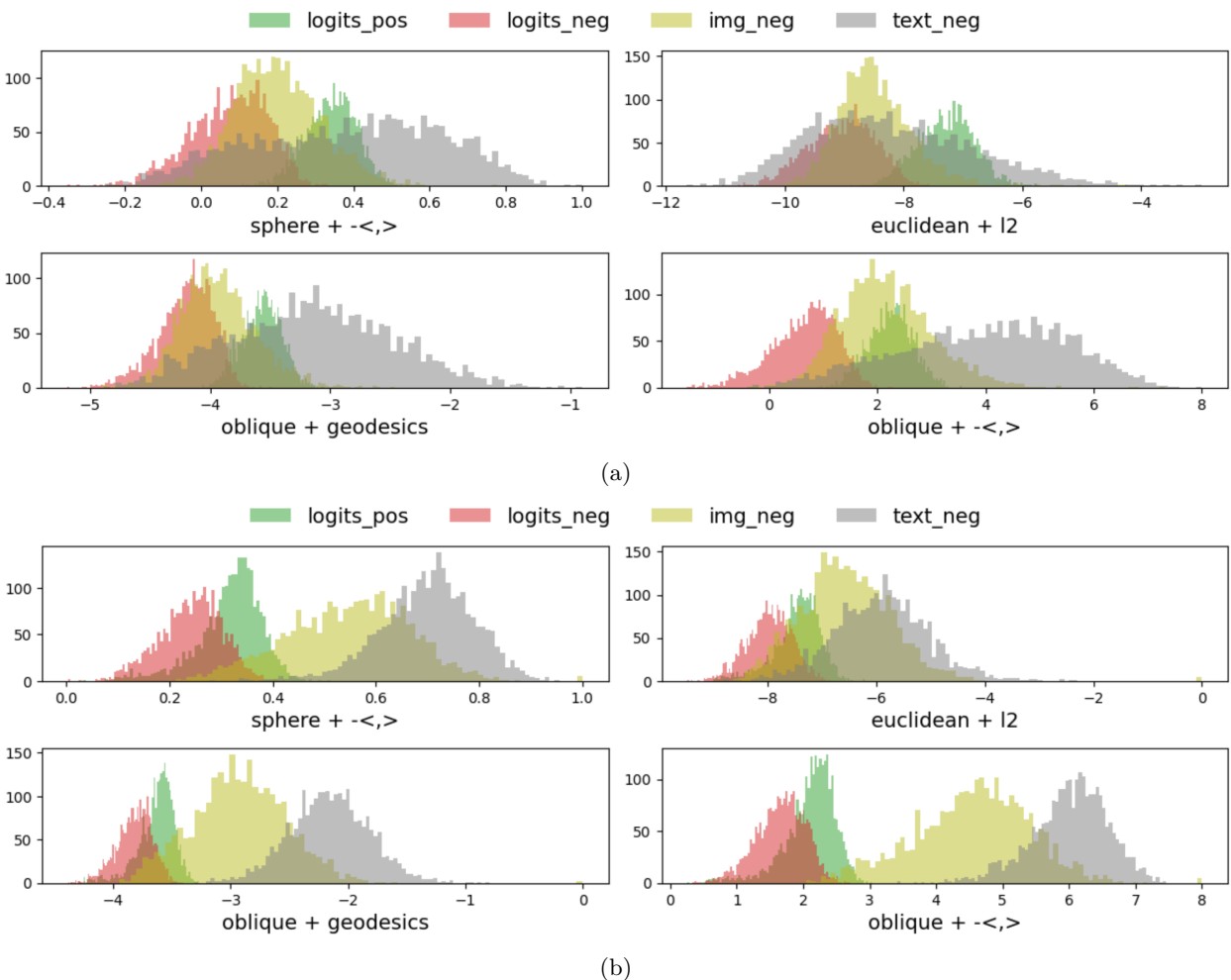

Figure 6: Visualization of the distribution of distances between samples. The `logits_pos` and `logits_neg` denote the distances between positive and negative image-text pairs, respectively. The `img_neg` and `text_neg` denote the distances between negative image-image and text-text pairs, respectively. The models are trained using the Yfcc datasets, (a) and (b) depict the distribution of in-domain data (Yfcc) and out-domain data (RedCaps), respectively.

## A.4 Additional Ablation on Oblique Structure

We provide more ablation results regarding the structure of the oblique manifold under fixed total dimensions in Table 8. We can observe that the Ob(32, 32) configuration performs the best in general, while the sphere with more 1024-dimensional embedding has slightly better linear probe performance. We also notice that a more complicated structure provides better text-to-image retrieval results.

## A.5 Additional Results on the ECCV Dataset

We provide more results using the ECCV dataset Chun et al. (2022). The dataset is proposed for eliminating the false negatives samples in the validation set of the original MSCOCO dataset. Instead of the commonly used Recall@K (R@K) metric, the datasets provide a new ranking-based metric mAP@R. The authors of the ECCV dataset have shown that the mAP@R metric is more aligned to humans than Recall@k. Therefore, the performance of a model evaluated by mAP@R would be less occasional than the R@1. We employ the

| Method | COCO 1K | | COCO 5K | | CxC | | ECCV Caption | | | | | |
| *baseline[impl.]* | **I2T** | **T2I** | **I2T** | **T2I** | **I2T** | **T2I** | | **I2T** | | | **T2I** | |
| | R@1 | R@1 | R@1 | R@1 | R@1 | R@1 | mAP@R | R-P | R@1 | mAP@R | R-P | R@1 |
| *ViT-B/16-224 as visual bone.* | | | | | | | | | | | | |
| CLIP[openAI†] | 71.7 | 52.5 | 52.5 | 33.1 | 54.0 | 34.7 | 23.7 | 34.0 | 68.8 | 34.8 | 44.0 | 73.4 |
| CLIP[openCLIP‡] | 74.0 | 57.6 | 55.4 | 38.3 | 57.3 | 40.0 | 26.2 | 36.6 | 70.3 | 36.9 | 46.4 | 77.5 |
| CLIP[our-impl.] | 80.8 | 63.0 | **64.2** | **43.1** | **65.3** | **44.9** | 30.5 | 41.0 | **78.6** | 40.5 | 49.9 | 81.2 |
| CLIP[Multi(32,16)] | **81.1** | **63.1** | 63.8 | 42.9 | **65.3** | 44.8 | **30.9** | **41.7** | 76.3 | **41.7** | **50.5** | **84.1** |
| *ViT-L/14-224 as visual bone for reference.* | | | | | | | | | | | | |
| CLIP[openAI†] | 74.3 | 55.4 | 56.4 | 36.6 | 58.0 | 38.3 | 24.0 | 33.8 | 71.3 | 32.0 | 41.8 | 73.0 |
| CLIP[openCLIP‡] | 77.2 | 61.4 | 59.7 | 43.0 | 61.1 | 44.8 | 28.1 | 38.3 | 73.0 | 38.7 | 47.9 | 81.2 |

Table 7: Comparsion of large scale contrastive visual-textual pre-train model on benchmark datasets.

| Topology | Distance | Zero-Shot I2T R@1 | Zero-Shot T2I R@1 | Zero-Shot Cls. Acc. | Linear Probe Cls. Acc. |
| --- | --- | --- | --- | --- | --- |
| Temperature, init=$e^{2.64}$, gradient=True | | | | | |
| Sphere(512) | $-\boldsymbol{u}^T\boldsymbol{v}$ | 48.3 | 31.45 | 30.62 | 60.38 |
| Sphere(1024) | $-\boldsymbol{u}^T\boldsymbol{v}$ | 50.7 | 32.05 | 29.60 | **60.53** |
| Ob(128, 8) | $-\mathrm{tr}(\boldsymbol{u}^T\boldsymbol{v})$ | 49.4 | 32.85 | 30.55 | 60.12 |
| Ob(64, 16) | $-\mathrm{tr}(\boldsymbol{u}^T\boldsymbol{v})$ | 50.3 | 33.25 | 30.34 | 60.16 |
| Ob(32, 32) | $-\mathrm{tr}(\boldsymbol{u}^T\boldsymbol{v})$ | **52.3** | **33.47** | **30.62** | 60.32 |

Table 8: The retrieval and classification performance of the proposed approach using different oblique manifold structures and the multi-token implementation. "gradient={True/False}" donates if the temperature is learnable.

officially released evaluation tool and summarize the performance of the models in Table 7. It is clear that our proposed multi-token oblique topology has better performance under the mAP@R metric.

## A.6 Additional Results Using the TCL Framework

We combine our proposed method with the TCL model Yang et al. (2022), which is one of the state-of-the-art vision-language retrieval models that employ contrastive visual-textual alignment in its earlier stage. During the pre-training, the TCL induces a mixture of in-modal and cross-modal contrastive losses, while conducting the masked language modeling (MLM) and image-text matching tasks simultaneously. During the testing, the cross-modal contrastive alignment head first lists sample pairs with high similarity scores, then these pairs are fed into the matching head to obtain the final matching scores. We alternate the topologies of all the embedding spaces with Ob(128,2); more precisely, we change the normalization function as shown in Section 3.1. For the experimental analysis in this subsection, we follow the configurations of the reference models, employ a collection of CC3M (Sharma et al., 2018), MSCOCO Captions (Chen et al., 2015), Visual genome (Krishna et al., 2017) and SBU (Ordonez et al., 2011) as the pre-training dataset, which contains roughly 4 million annotated image-text pairs. The models are then evaluated using Flickr30k (Plummer et al., 2015) and MSCOCO Captions (Chen et al., 2015).

The results are shown in Table 9. Since our method does not affect the matching head, we also report the performance of the contrastive alignment head. In general, our method improves the average recall performance, but the improvement is not significant. We consider the reasons as i) The method (or recent similar methods) employs pre-trained vision and language models, as well as a matching head and an MLM head; hence it is less sensitive to the gradients from the contrastive alignment; ii) The datasets employed for training contain less noise, while the training is scheduled with an overlength scheme (the zero-shot performance does not increase in the last 5 epochs).

**Additional Notes on TCL** We also provide the comparison results with officially released checkpoints. It can be seen that our implementation performs 0.5-1.0% worse than the official checkpoints. On the other

| Method
*baseline[impl.]* | Flickr | | | Coco | | |
|---|---|---|---|---|---|---|
| | **I2T**
R@1 | **T2I**
R@1 | **Recall**
mean | **I2T**
R@1 | **T2I**
R@1 | **Recall**
mean |
| *Zero-shot performance.* | | | | | | |
| TCL[official] | 93.00 | 79.60 | 93.97 | 71.40 | 53.50 | 79.49 |
| | (84.20) | (67.10) | (88.45) | (55.40) | (40.80) | (69.92) |
| TCL[our-impl.] | 91.00 | 78.28 | 93.25 | 70.16 | 53.05 | 79.07 |
| | (83.30) | (68.40) | (88.73) | (57.34) | (43.21) | (71.31) |
| TCL[Ob(128,2)] | 91.20 | 78.14 | **93.29** | 70.14 | 53.35 | **79.14** |
| | (84.80) | (67.86) | (88.84) | (57.10) | (43.13) | (71.32) |
| *Fine-tuned performance.* | | | | | | |
| TCL[official] | 94.90 | 84.00 | 95.57 | 75.60 | 59.00 | 82.87 |
| | (87.90) | (71.38) | (90.92) | (65.34) | (48.94) | (76.53) |
| TCL[our-impl.] | 93.80 | 83.06 | 95.17 | 73.56 | 57.74 | 82.06 |
| | (88.30) | (72.94) | (91.27) | (66.98) | (50.34) | (77.43) |
| TCL[Ob(128,2)] | 93.80 | 82.90 | **95.18** | 74.78 | 57.72 | **82.13** |
| | (88.60) | (73.26) | (91.39) | (65.60) | (49.83) | (76.86) |

Table 9: Retrieval performance on Flickr30K and MSCOCO of our implemented TCL model and the variant using our proposed method. The numbers in brackets are the performance obtained using the contrastive alignment head.

| #Tokens | 1 | 2 | 4 | 8 |
|---|---|---|---|---|
| **Top1 Acc.** | $13.0\pm9.7$ | $21.5\pm14.3$ | $27.7\pm20.4$ | $62.1\pm4.4$ |

Table 10: ImageNet zero-shot classification performance of CLIP[Multi(32,16)] model using a randomly selected subset of [CLS] tokens.

hand, our implementation has better alignment head performance. Since we are employing the codes released in the official repository, the reason might be the following: i) Datasets difference, that we have ∼3000 fewer images in the SBU dataset while owning 5000 more images in the CC3M dataset; ii) We resize the CC3M dataset to short edge 500 pixels, while the official repository does not clearly provide the pre-processing approach; iii) We implicitly have a short training time or smaller matching loss weight than the official checkpoints due to the difference in the framework.

## A.7  Test of Mixture-of-Expert Hypothesis:

We investigate the mixture-of-expert hypothesis of the proposed method. Since the [CLS] token is considered to encode the global representation of the sample, the employment of multiple [CLS] tokens may function in a mixture-of-expert style. That is, after training, each sub-sphere (or a subset of sub-spheres) in the oblique structure is capable of alignment. Then, the system functions as a mixture of weak alignment models (experts). To test this hypothesis, we calculate the zero-shot classification performance of the CLIP[Multi(32,16)] model with randomly selected subsets of sub-spheres. From Table 10, we find that the drop in performance is reasonably small (∼12%) with half of the alignment tokens. This result reveals a possible mechanism of the oblique structure during optimization, where a subset of sub-spheres is priorly aligned.

## A.8  More Visualization using GradCAM

In Figure 7, we provide more visualization results using GradCAM.

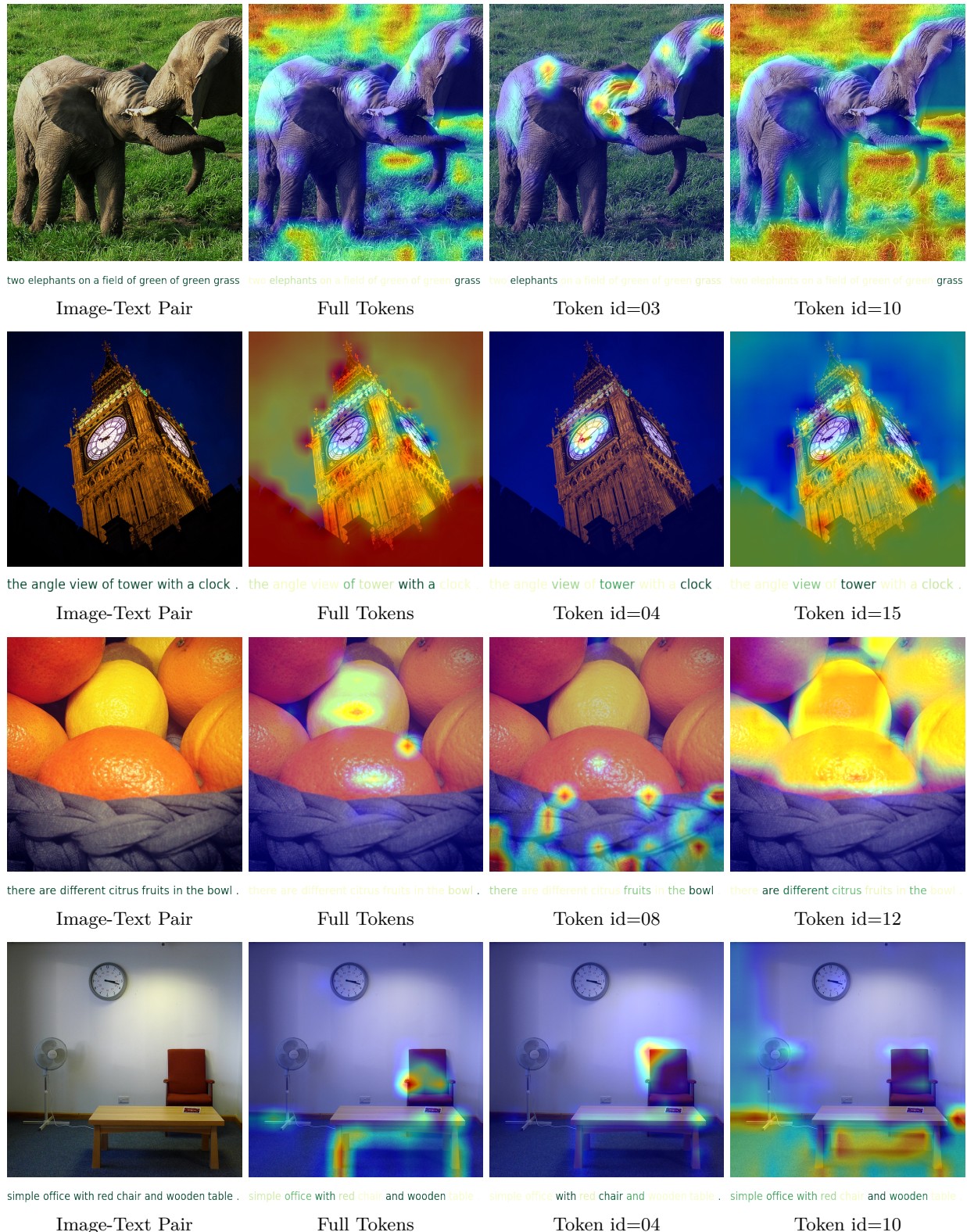

Figure 7: More visualization of the importance map using the Grad-CAM algorithm. See Section 4.5 for details.

### A.9 A note on the recent advance in noisy image-text matching

Recently, many pieces of research have been made to tackle the noisy image-text matching problem in contrast learning. Below, we provide a concise survey of these works, for the readers who want to know more about this topic. Although these works may not be comparable with our proposed method, they still support that the noisy visual-textual correspondences is an important research topic in this field.

**Chun et al. (2022)**: This paper argues that existing ITM benchmarks have a significant limitation of many missing correspondences. Then it proposes a new dataset, ECCV Caption, to correct the massive false negatives and proposes a new metric, mAP@R, to evaluate VL models.

**Li et al. (2023)**: This paper proposes a method to correct false negatives by integrating language guidance into the ITM framework. This framework corrects the locations of false negatives in the embedding space.

**Chun (2023)**: This paper also argues that the image-text matching task suffers from ambiguity due to multiplicity and imperfect annotations. Then, this paper proposes an improved probabilistic ITM approach that introduces a new probabilistic distance with a closed-form solution.

**Huang et al. (2021)**: This paper points out that the training data may contain mismatched pairs. To learn the noisy correspondence, the authors divide the data into clean and noisy partitions and then rectifies the correspondence via an adaptive prediction model.

**Qin et al. (2022)**: This paper considers the major challenge in cross-modal retrieval is the noisy correspondence in training data. This refers to the fact that some of the training pairs may not be correctly aligned, *i.e.*, the image and text do not actually correspond to each other. They propose a framework to address this challenge by integrating two novel techniques: Cross-modal Evidential Learning and Robust Dynamic Hinge.

**Yang et al. (2023)**: This paper proposes a general framework for cross-modal matching that can be easily integrated into existing models and improve their robustness against noisy data. This framework estimates soft labels for noisy data pairs by exploiting the consistency of cross-modal similarities.

**Han et al. (2023)**: The paper proposes a Meta Similarity Correction Network to provide reliable similarity scores for cross-modal retrieval. The method learns to distinguish between positive and negative pairs of data using meta-data, and can be used to remove noisy samples from the training dataset.

