# OpenReview forum: "Design of the topology for contrastive visual-textual alignment"
_TMLR — Rejected by TMLR_

### Review · Reviewer_XRy3 · 2023-07-22

**Summary Of Contributions:**

This manuscript investigates the impact of learnable softmax temperature on contrastive visual alignment, analyzes several properties of the embedding space and implements the objective topology with multiple class tokens of the transformer architecture.

**Audience:**

No

**Claims And Evidence:**

Yes

**Requested Changes:**

1. Please check the typos, such as on page 3 in the ‘Visual-Textual Pre-trained Model’ section, line 3, ‘fθsimultaneously’. And on line 6, it should be ‘extracted from.’Additionally, on page 12, there is a typo, ‘limiation’, and on page 17, ‘tempearture’ should be corrected.
2. Please ensure consistency in the naming of the dataset throughout the paper. A widely accepted convention would be to use 'Flickr30K' and 'MSCOCO' as the more commonly recognized names. However, on page 10, in section 4.5, they are referred to as 'Flicker and MSCoco datasets'. Please revise this to maintain uniformity in the nomenclature.
3. Please ensure accurate usage of specialized terminology throughout the paper. In section 4.2, on the fourth line, "visual bone" should be revised to "visual backbone" to accurately represent the intended concept.
4. I notice in the appendix that 32 V100 training blocks took one day. How do you achieve this much faster than the original CLIP training.


**Strengths And Weaknesses:**

Strengths:
The manuscript provides a solid foundation by using the oblique manifold as the embedding space and employing the (negative) inner product as its distance function. It goes on to compare the theoretical characteristics of various topological structures, accompanied by extensive experimental findings and detailed analysis for each hyperparameter.
Weaknesses:
1.The method proposed in this manuscript appears to be relatively simple and lacks innovation. It primarily combines the concept of oblique manifold in Riemannian geometry with contrastive learning. While the results have shown improvement, it does not explicitly establish the direct relationship between this approach and resolving the issue of semantic ambiguity in contrastive learning.
2. The explanation of the ‘Relaxed’ triangular inequality in the proposed method is unclear. According to the given scenario, if the model is ‘well optimized’, then even in the presence of semantic relevance, d(u*1, v*2) should be greater than d(u*2, v*2), indicating that the right side should not be 3 ϵ+. Moreover, this inequality itself is merely a mathematical expansion and does not have a direct connection with solving the problem of semantic ambiguity. By adopting different spatial strategies, all the distances on both sides of the inequality will be simultaneously amplified, which may not align well with the proposed motivation.

---

### Review · Reviewer_WRo6 · 2023-07-29

**Summary Of Contributions:**

This paper proposes a new similarity function design for CLIP-like visual-textual contrastive learning approaches. To be specific, this paper employs the "oblique manifold", which is not well-described in this paper (I will mention it in the next section). The implementation of a metric function for this oblique manifold is done by using multiple [CLS] tokens. Using this approach, the proposed similarity function shows better performance than the cosine similarity function in the YFCC-15M CLIP training experiment.

**Audience:**

Yes

**Broader Impact Concerns:**

As my previous comment, I found that this paper copies the explanation of the oblique manifold from https://www.manopt.org/manifold_documentation_oblique.html

After I had found this, I also checked whether there is any other non-cited but copied texts. Even though I couldn't find any, if there is any, the authors should re-write the sentences or should cite them and clarify that the words are copied from the reference. At least, I think the main contribution of this paper is not plagiarism, I don't think it is a serious issue as much as academic plagiarism. However, in terms of research ethics, it would be better to clarify the reference or fix the copied sentence.

**Claims And Evidence:**

Yes

**Requested Changes:**

Overall, I think this paper can be interesting to some TMLR readers. However, I found many significant flaws in this paper, mostly in the presentation. I think the current version of this paper is not acceptable, as it is hard to understand and many details are not sufficiently provided. I think if the presentation of this paper can be improved, this paper would be worth being accepted at the TMLR, as the topic and the main idea of this paper look intriguing.

## Presentation

The most significant drawback of this paper lies in the presentation. As this is a journal paper, and there is another change to improve the text, I would like to suggest a heavy revision with the following items.

1. Please describe more about the oblique manifold. Especially, what is the meaning of the oblique manifold, what is the advantage of the oblique manifold compared to other Riemannian manifolds, such as Euclidean space and a sphere manifold.
2. The description of the oblique manifold should be re-written by the authors, or please cite https://www.manopt.org/manifold_documentation_oblique.html because it is a totally copied paragraph from the web site.
3. Please make Sec 3.1. contain more details. I still cannot get the difference between single-token and multi-token implementations. I strongly suggest adding illustrations of the oblique implementations and other similarity functions.
4. Please provide more implementation details of the proposed similarity function. There is a python pseudo-code, but I don't think it is a good idea to just put the whole code without a precise explanation of the implementation. Especially, please provide the rationale beyond the hyperparameter selection. I totally understand that it is not possible to explore every possible combination of the hyperparameters, but at least, I expect minimal explanations of the rationale of the chosen values.
5. 1-4 will require more pages. I don't think Figure 1 and 3 are sufficiently informative compared to 1-4. Please consider moving Figure 1, 3 to Appendix to make more room for 1-4.

## Technical ambiguities

Please fix the following items. If they are not wrong, please consider providing more details of each item

- Weakness: Ambiguity in the proposed method?
- Weakness: Technical flaws, or overclaim in Section 3.2 (i) memory consumption
- Weakness: Ambiguous statement in Section 3.2 (iii) triangular inequality
- Weakness: Minor comments

## Related works for false positives

I recommend adding "ECCV Caption: Correcting False Negatives by Collecting Machine-and-Human-verified Image-Caption Associations for MS-COCO" paper for justifying the false positive argument in Introduction. This paper shows that there are actually many false positives in a popular VL dataset. Also, I found another related works focusing on the false positive issue: "Integrating Language Guidance into Image-Text Matching for Correcting False Negatives" and "Improved Probabilistic Image-Text Representations", that show that false positives can lead to a wrong learned metric space. I don't think the latter papers can be directly compared with the proposed method. I also don't think it is necessary to cite those papers, but citing those papers can improve the justification of the false positive argument. I also recommend trying the ECCV Caption evaluation in a zero-shot manner (but not a mandatory, the authors can ignore this comment if they don't think it will make the submission stronger)

- Chun, Sanghyuk, et al. "Eccv caption: Correcting false negatives by collecting machine-and-human-verified image-caption associations for ms-coco." European Conference on Computer Vision. Cham: Springer Nature Switzerland, 2022.
- Li, Zheng, et al. "Integrating Language Guidance into Image-Text Matching for Correcting False Negatives." IEEE Transactions on Multimedia (2023).
- Chun, Sanghyuk. "Improved Probabilistic Image-Text Representations." arXiv preprint arXiv:2305.18171 (2023).

I also found other related works arguing about the noisy image-text correspondences that mainly tackle the problem when the correspondences between images and texts are noisy. It is not 100% likely to the target task, but it would be helpful to cite them for supporting that many studies tried to tackle the noisy VL correspondences.

- Huang, Zhenyu, et al. "Learning with noisy correspondence for cross-modal matching." Advances in Neural Information Processing Systems 34 (2021): 29406-29419.
- Qin, Yang, et al. "Deep evidential learning with noisy correspondence for cross-modal retrieval." Proceedings of the 30th ACM International Conference on Multimedia. 2022.
- Yang, Shuo, et al. "BiCro: Noisy Correspondence Rectification for Multi-modality Data via Bi-directional Cross-modal Similarity Consistency." Proceedings of the IEEE/CVF Conference on Computer Vision and Pattern Recognition. 2023.
- Han, Haochen, et al. "Noisy Correspondence Learning with Meta Similarity Correction." Proceedings of the IEEE/CVF Conference on Computer Vision and Pattern Recognition. 2023.

## Typos and wrong expressions

- Pg. 3: LaTex error in "simultaneously"
- Pg. 3: the oblique manifold Ob(n, n) -> Ob(n, m)
- Please replace "Coco" to "COCO". Also, COCO is not introduced in the "Datasets" paragraph. Please add it.

**Strengths And Weaknesses:**

## Strengths

- The proposed method looks well-working.
- Applying a new manifold to the vision-language domain can be intriguing to some TMLR's audience.

## Weakness

### Not enough description of the proposed method

The most serious problem of this paper is that the description of the proposed method is extremely limited. I only get very limited information on (1) what is the oblique manifold, (2) how the proposed similarity function is designed, and (3) how the proposed multi-token oblique implementation is a similarity function on an oblique manifold. I have put a lot of effort into understanding the context, but I couldn't sufficiently understand what the method is and how the method works.

The definition of the oblique manifold is only briefly described in page 3. Moreover, I found that the description is identical to the description from https://www.manopt.org/manifold_documentation_oblique.html
A novice reader might wonder what is the advantage of the oblique manifold compared to other Riemannian manifolds, such as Euclidean space and a sphere manifold.

Also, Section 3.1 only contains very limited information about the method. I cannot fully understand how to compute the proposed similarity only with 3.1. For example, as far as the reviewer understood, this paper tried to map an input to not a Euclidean space (or a sphere manifold), but to the set of multiple sphere manifolds. The similarity between two inputs is computed by tr(u^T v), but there is no explanation of how and why tr(u^T v) can be a metric function in an oblique manifold.

Finally, the token-based oblique implementation can be more detailed. One can illustrate the detailed computation process with a simple diagram for the proposed similarity and other similarity functions in Table 1.

### Ambiguity in the proposed method?

Given the limited information, the reviewer presumes that the proposed token-based oblique implementation would be wrong. Following the definition of an oblique manifold, the column vectors should satisfy that $diag(u^T u) = I$. However, it seems that the implementation only applies "L2-normalization" for each column. Unless all column unit vectors are the same, as far as the reviewer understood, $diag(u^T u)$ cannot be $I$, as $u_i^T u_j = \| u_i \| \| u_j \| \cos \theta = \cos \theta$ where $u_i$ denotes i-th unit column vector and $\theta$ denotes the "angle" between i, j unit vectors. I think that U and V should be transformed by an orthonormalization process, such as Gram–Schmidt process or QR decomposition, rather than a column-wise l2 normalization. Please correct me if I am wrong.

### Technical flaws, or overclaim in Section 3.2 (i) memory consumption

If the reviewer understood correctly, the memory resource assumption in Table 1 is wrong. In Section 3.2 (i), this paper argues that
> "For instance, given a mini-batch of sample pairs of size b with d−dimensional output, the computation of the inner product achieves a complexity of O(b^2 d) and storage usage of O(b^2). However, since the back-propagation of the ℓ2−norm requires intermediate results that cannot be “inplace” calculated, the ℓ2−norm (or any ℓp−norm based distance) in euclidean space requires a storage usage of O(b^2d)."

However, as the backpropagation of the inner product needs the input vector itself (it is because $\frac{\partial (u^T v)}{\partial u} = v$), regardless of the "in-place" operation, it needs the same storage to the l2 distance counterpart.

If this text means that the memory consumption "purely" by the similarity function (i.e., the additional memory consumption for saving partial derivatives of the computational graph of the similarity function), I would like to say that it is almost neglectable compared to that of the other computational graphs. For example, assume the ViT-B/16 CLIP backbone, then the total parameter size is 150M. Roughly speaking, we probably need to save more than (batch size * hundreds of millions) parameters for backpropagation (it actually depends on the input resolution and intermediate feature dimensions). On the other hand, the similarity backpropagation memory only needs "batch size" parameters, or even in the worst case, "batch size * output dimension (512 in this paper)". It is almost neglectable compared to the other memory resource of the full computational graph. I don't think the contribution of the memory resource is meaningful. Table 1 can lead to a misunderstanding that the "Sphere topology" consumes "1/d times" smaller memory than "Euclidean topology", which is completely wrong.

### Ambiguous statement in Section 3.2 (iii) triangular inequality

I cannot understand how Eq. (4) is derived. First, it is not about the triplet points but quadruple points. When I tried to derive (4), I noticed that there is no information on d(v1, v2) or d(u1, u2). If the reviewer understood correctly, the triangular inequality cannot be derived without d(v1, v2) or d(u1, u2).

Second, I cannot get why d(u1, v2) is considered a positive pair here. I presume that the only positives are (u1, v1) and (u2, v2), while (u1, v2) and (u2, v1) are "false positives", i.e., they are annotated to "negatives" in the dataset, but they are actually "positives". If this is correct, I cannot understand why d(u1, v2) is considered a positive by a "well-optimized" model. If the paper would like to argue that the strict triangular inequality could lead to a broken metric space due to false positives, yes, I agree with the argument, but I am not sure what this paragraph wants to describe.

### Not enough (or confusing) implementation details

Not only the description of the proposed method is ambiguous, but also the implementation details are somewhat not sufficient. For example, I cannot get how many [CLS] tokens are appended for the multi-token oblique implementation. As far as the reviewer understood, the multi-token oblique implementation repeats the [CLS] tokens multiple times, e.g., if the number of [CLS] tokens is 4, then the total output dimension becomes 4 * 512 = 2048. It could be critical in terms of the computational resource, or batch size. However, it seems that the implementations in the experiments looks the single-token oblique implementation because 16 * 32 = 512 and 64 * 8 = 512. However, it denotes "Multi(16, 32)" where Multi(n, m) is only defined in "multi-token oblique implementation", which is totally confusing.

### Minor comments

- There is no justification for the choice of n, m for Ob(n, m). For example, Why Table 2 uses 16, 32 and Table 3 uses 64, 8?
- What is "Recall mean" in Table 2? I presume it is the average of R1, R5, R10 of i2t, t2i retrieval at the first time, but the values are smaller than my first assumption. Please clarify the metric.
- I don't get why the mixture-of-expert hypothesis pops up on pg 10. It first introduces here without any previous discussion. It should be discussed previously or should be discussed what the mixture-of-expert hypothesis means in this context.
- It is not a weakness, but there are some related works that can support false positive arguments. See the next section for details.

---

### Review · Reviewer_bSxW · 2023-08-05

**Summary Of Contributions:**

This work proposes an improved version of CLIP training, which yields a better model that achieves significantly higher performance on zero-shot image classification tasks than the original CLIP model. The authors mainly investigated how the topology of the text-visual embedding space can be alternated to further the alignment of the text-visual embeddings. This paper also shows that trainable temperature in the contrastive learning loss effectively yields a distance scaling factor. The authors continued to argue that the proposed topology is a better solution than this learnable temperature. Finally, the paper shows many ablation studies supporting their claim that the proposed multi-token oblique method works better than the original and other alternative methods.

**Audience:**

Yes

**Broader Impact Concerns:**

No clear negative society impacts from this.

**Claims And Evidence:**

Yes

**Requested Changes:**

See the weakness.

**Strengths And Weaknesses:**

Strengths:
The multi-token CLIP achieves significantly better results than the original CLIP algorithm. The authors also carefully reproduced the reported CLIP algorithm on the training dataset they collected, which means that the improvement is due to the algorithm improvement instead of the data difference. The ablation studies in the paper also show that this improvement is general to architecture and is not very sensitive to hyperparameter settings. The explanation of the effect of the trainable temperature in contrastive learning loss is also convincing.


Weaknesses:
One technical detail on how the multi-token method is implemented is unclear. From the notation (16, 32) and the projection dimension values in the appendix, it seems that the authors reduced the projection dimension from the original 512 value to the new 32 value. But does this also mean that the embedding dimension for the other visual tokens is also changed to 32? I would think so, as otherwise the CLS token cannot be simply computed from the other visual tokens. But this seems to be a big architectural change that would make the number of trainable parameters and computation efficiency significantly reduced, which the authors need to confirm. In fact, why do the authors need to reduce the number of projecting dimensions? If it’s just about adding more CLS tokens, why cannot the authors just add 15 more CLS tokens of 512 dimension?

I don't think the authors released their source code and pretrained models. It would be great for these to be shared as well.

---

### Decision · Action_Editor_fYHq · 2023-10-23

**Recommendation:** Reject

**Comment:**

This submission proposes to construct image and text embeddings that lie on the oblique manifold, in contrast to the hyperspherical manifold used in conventional contrastive image/text approaches. The proposed approach involves two changes to the contrastive image/text model. First, it uses multiple CLS tokens instead of just one. Second, whereas CLIP projects the representation of this CLS token, L2 normalizes it, and then takes an inner product, the proposed approach instead projects each CLS token, L2 normalizes them separately, and then takes the sum of their inner products. This approach appears to provide a meaningful over CLIP baselines from previous work and a CLIP baseline implemented by the authors, across many standard tasks. Most notably, the proposed method achieves a 7% improvement in ImageNet zero-shot classification accuracy, which is non-trivial and well outside random variability.

Reviewers felt that the improvements over CLIP were convincingly demonstrated. However, they initially raised concerns regarding the presentation, particularly in the description of the oblique manifold and its relevance to the proposed technique. Although reviewers felt that the authors' revisions improved the paper, their recommendations indicate that they still find the presentation of the oblique manifold unsatisfying.

Like the reviewers, I feel that the presentation of the technique could be clearer, but the submission contains sufficient information to understand what was done. I am more concerned that some of the claims regarding the relationship between the oblique manifold and the observed results do not appear to be well-substantiated. In Section 4.3, the results do not provide convincing evidence for the three claims made. It is true that geodesic distance consistently outperforms Euclidean distance and the inner product distance outperforms the geodesic distance more often than not. However, it is a stretch to directly link these differences to uniformity and adherence to the triangle inequality, respectively, given that these are not the only differences among these distance measures. The discussion of the triangle inequality in Section 3.2 relates the importance of the "relaxed" triangle inequality to the existence of false negative pairs, but no experiments substantiate this claim.

Beyond the issues with the claims above, I'm not fully convinced that it is oblique topology and not other differences between methods that drive the improvements in Table 2. Of the two modifications made here (multi-token and use of oblique distance), the vast majority of discussion involves the latter. Table 4 performs some ablations to indicate that both modifications provide improvement, but this is not entirely clear-cut, because the improvements are smaller. In particular, among the image classification results, only the 31.93% number would be a statistically significant improvement over the 30.62% baseline assuming the normal variance on ImageNet. (I assume this is ImageNet, but I can't find where the datasets in Table 3 and Table 4 are stated.) To make these experiments more convincing, I'd suggest (1) providing some measure of run-to-run variability in Table 4 (standard deviation, standard error, or a 95% confidence interval) and (2) providing a baseline that uses multiple CLS tokens but simply concatenates their projected representations and uses cosine similarity in the loss. If the authors have the compute, the most convincing thing to do would be to ablate multi-token and use of the oblique distance separately in the full setting in Table 2, but I realize that may not be possible.

Overall, this submission improves convincingly over CLIP, but there are too many claims that fail to meet TMLR's standard of "accurate, convincing and clear evidence" for me to be able to accept it in its current form. I thus recommend a major revision.

**Audience:**

Yes.

**Claims And Evidence:**

The claim that the proposed approach improves performance is well-substantiated. However, the relationship between this performance improvement and the properties of the oblique manifold is not well-substantiated. See further discussion under below.

**Resubmission Of Major Revision:**

The authors may consider submitting a major revision at a later time.